# Multi-level Correspondence via Graph Kernels for Editing Vector Graphics Designs

Hijung V. Shin*
Adobe Research

Jeremy Warner†
University of California, Berkeley

Björn Hartmann‡
University of California, Berkeley

Celso Gomes§
Adobe Research

Holger Winnemöller¶
Adobe Research

Wilmot Li‖
Adobe Research

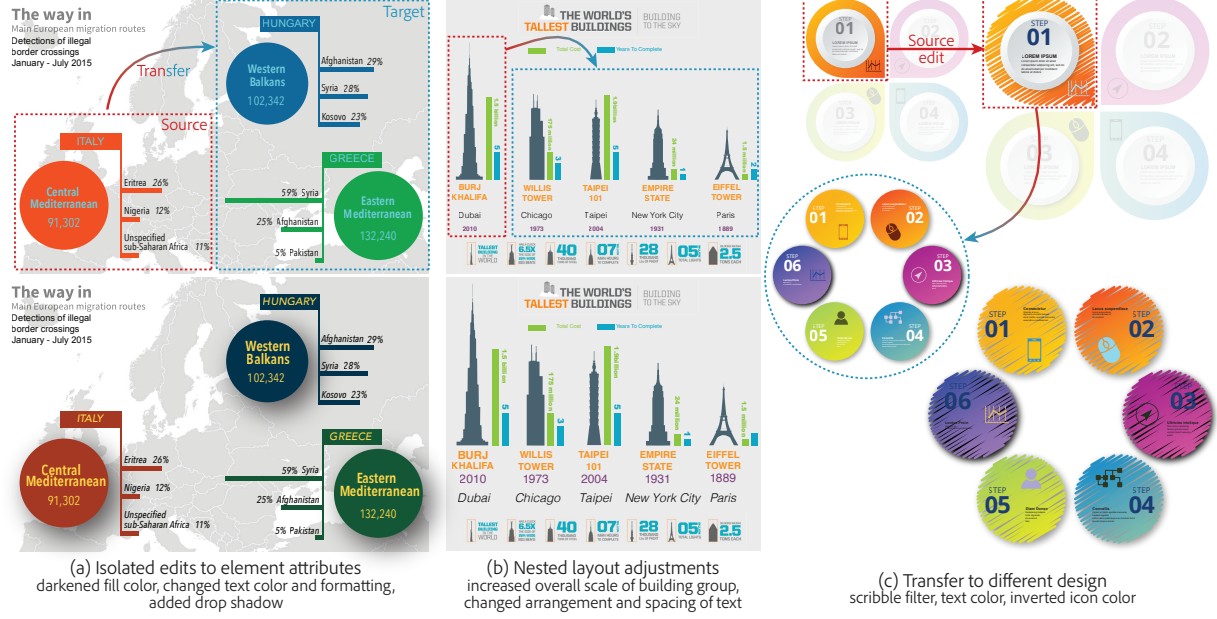

(a) Isolated edits to element attributes
darkened fill color, changed text color and formatting,
added drop shadow

(b) Nested layout adjustments
increased overall scale of building group,
changed arrangement and spacing of text

(c) Transfer to different design
scribble filter, text color, inverted icon color

Figure 1: Graphic designs often contain repeating sets of elements with a similar structure. We introduce an algorithm that automatically computes this shared structure which enables graphical edits to be transferred from a set of source elements to multiple targets. For example, designers may want to propagate isolated edits to element attributes (a), apply nested layout adjustments (b), or transfer edits across different designs (c).

## ABSTRACT

To create graphic designs such as infographics, UI mockups, or explanatory diagrams, designers often need to apply consistent edits across similar groups of elements which is a tedious task to perform manually. One solution is to explicitly specify the structure of the design upfront and leverage it to transfer edits across elements that share the predefined structure. However, defining such a structure requires a lot of forethought which conflicts with the iterative workflow of designers. We propose a different approach where designers select an arbitrary set of *source* elements, apply the desired edits, and automatically transfer the edits to similarly structured *target* elements. To this end, we present a graph kernel-based algorithm that retroactively infers the shared structure and correspondence between source and target elements. Our method does not require any explicit annotation and can be applied to any existing design regardless of how it was created. It is flexible enough to handle differences in structure and appearance between source and target graphics, such as cardinality, color, size, and arrangement. It also generalizes to different types of edits such as style transfer or applying animation effects. We evaluate our algorithm on a range of real-world designs and demonstrate how our approach can facilitate various editing scenarios.

**Index Terms:** Computing methodologies—Computer graphics—Graphics systems and interfaces—; Computing methodologies—Computer graphics—Graphics systems and interfaces—

## 1 INTRODUCTION

Graphic designs such as infographics, UI mockups, and explanatory diagrams often contain multiple sets of elements with similar visual structure. For example, in Figure Fig. 1(a) each chart is represented by a circle, a country name tag, three data bars, and text annotations arranged in a consistent manner. There are similar repetitions across the graphics for each building in Figure 1(b). In some cases, we also see elements with consistent visual structure across multiple different designs. For example, Figure 1(c) shows two separate diagrams created by the same designer that share a similar underlying structure.

Designers often need to apply consistent edits such as style

* e-mail: vshin@adobe.com
† e-mail: jeremy.warner@berkeley.edu
‡ e-mail: bjoern@eecs.berkeley.edu
§ e-mail: cegomes@adobe.com
¶ e-mail: hwinnemo@adobe.com
‖ e-mail: wilmotli@adobe.com

changes, layout adjustments, and animation effects across these repeating sets of elements. For example, Figure 1(a) shows adjustments to fill color and adding drop shadows for various elements, and Figure 1(b) shows modifications to the spacing and layout of the text and graphics for each building. Such edits are tedious to perform manually, especially as the number of elements increases. One solution is to explicitly specify the structure of the design upfront and leverage it to transfer edits across sets of elements that share the predefined structure. For example, Microsoft PowerPoint's master slide feature allows users to edit the appearance of multiple slides at once. Similarly, popular UX design tools (e.g., Adobe XD, Figma) encourage users to define master symbols or components that control the properties of repeated instances within a design, such as buttons, icons, or banners. Defining such a structure ahead of time requires substantial forethought which often conflicts with designers' workflows. In many cases, designers create and iterate on the whole graphic to get the overall design right before thinking about repeated elements, shared structure, or what edits to apply.

We propose a different approach to help designers apply edits consistently across a design. Instead of asking users to explicitly structure their content ahead of time, we allow them to select an arbitrary set of *source* elements, apply the desired edits, and then automatically transfer the edits to a collection of *target* elements. In this workflow, the system is responsible for retroactively inferring the shared structure between the source and target elements. Thus, our approach can be applied to any design, regardless of how it was created. Our method is not limited to graphics that share identical structures or elements but is flexible enough to accommodate common variations between the source and target graphics, such as color, shape, arrangement, or element cardinality. Moreover, it generalizes to different types of edits such as style transfer, layout adjustments, or applying animation effects.

A core challenge in realizing this approach is finding correspondences between the source and target graphics such that the appropriate source edits can be applied to each target element. First, both the source and target graphics may contain many similar elements, both related and unrelated. Second, the target graphics usually contain several differences from the source graphics. For instance, in Figure 1(a) the size, as well as the position of the data bars relative to the circle element, are slightly different from one another. The type and range of these differences vary for each design, making it hard to define a consistent matching algorithm based on heuristics. Finally, some types of edits must take into account nesting and ordering relationships between the elements. For example, in Figure 1(b), the designer may scale up the entire set of graphics for one of the buildings and then separately adjust the vertical spacing between the text elements. Transferring this edit properly to all the buildings requires that we identify the analogous hierarchical structure for the other graphical elements in the design by computing multi-level correspondence. Although designers often organize elements into various groups during the creation process, these are not reliable indicators of perceptual structure and do not always correspond to the desired hierarchy for an edit or transfer. Also, user-created groups typically do not encode the ordering of elements, which is important for temporal effects like animation.

The main contribution of our work is an automatic algorithm for determining the shared structure and correspondence between source and target elements that addresses these challenges. Our method is based on graph kernels. Given a source and target graphics, we compute *relationship graphs* that encode the structure of the elements. We then analyze the source and target relationship graphs using graph kernels to compute element-wise correspondences. We also introduce an efficient method to hierarchically cluster and sequence the target elements into ordered trees whose structure is consistent with the source graphics. Together, the correspondences and ordered trees make it possible to transfer edits from the source to target ele-

ments. We evaluate our algorithm on a range of real-world designs and demonstrate how our approach facilitates graphical editing.

## 2 RELATED WORK

**Inferring Structure in Graphical Designs**

A large body of work focuses on automatically estimating the internal structure of graphic designs to facilitate authoring and editing. For example, in the context of graphical patterns, Lun et al. compute perceptual grouping of discrete patterns [12], and Guerrero et al. encode the structure of a pattern in a directed graph representation to create design variations [5]. In the context of web designs, Kumar et al. leverage the tree structure of the DOM as well as its style and semantic attributes to create mappings between web pages [9]. Liu et al. generate semantic annotations for mobile app UIs by extracting patterns from the source code [11]. In the context of layout optimization, GACA automatically decomposes a 2D layout into multiple 1D groups to perform group-aware layout arrangement [26]. Other examples include structural analysis of architectural drawings [15], procedurally generated designs [18, 24], 3D designs [21] and 3D scenes [4, 23]. Our work focuses on 2D vector graphics designs, with the purpose of facilitating edit operations.

While some design software such as Figma [8] allows users to create graphical designs procedurally (which are then perfectly matched, edited, and animated), such tools are not commonplace, and the resulting designs lose their structural information if they are exported to portable file formats such as SVG or PDF. We propose a generic solution that only depends on the graphics, regardless of how they were created.

A different approach to purely automatic inference is mixed-initiative methods that take advantage of user interactions to infer structure. For example, previous work has analyzed user edits to extract implicit groupings of vector graphic objects [19], propagate fills and strokes in planar maps [1], detect related elements in slide decks [3], and infer graphical editing macros by example based on inter-element relationships [10]. Some existing techniques introduce new interactive tools and widgets that facilitate manipulation or selection of multiple elements [7, 25], or sub-parts of elements that are perceptually related [2] through a combination of explicit user actions and automatic inference. In contrast, we propose an automatic algorithm to determine the shared structure within graphic designs. Our method does not require user annotations or edit history, which means it can be applied to any existing vector graphics design, regardless of how it was created.

**Computing Correspondence between Graphic Designs**

Computing the correspondence between two analogous designs is a long standing problem in graphics with many applications. Many techniques have been developed for computing correspondences between a pair of images [20], 3D shapes [22], and 3D scenes [4]. These algorithms exploit local features as well global structures to compute the correspondence.

One technique that has proven highly effective for comparing different objects is kernel-based methods. There is ample work on defining kernels between highly structured data types [17]. In particular, one approach is to represent objects or collections of objects as a graph and define a kernel over the graphs. This approach has been applied to a variety of problems such as molecule classification [13], computing document similarity [16], and image classification [6]. Our algorithm is directly inspired by Fisher et. al's 3D scene comparison method [4], which uses graph kernels to compute a similarity between 3D scenes. To the best of our knowledge, there is no prior work that computes a pairwise element-to-element correspondence between two sets of vector-based graphical elements. In addition to element-wise correspondence, we also infer the nesting and ordering relationships between the elements, which is crucial for transferring complex edit operations.

## 3 OVERVIEW

The input to our method is a set of source elements that the user has manually edited and a set of target elements to which the user wants to transfer the edits. Transferring isolated changes to element attributes (e.g., fill color, text formatting) simply requires matching each target element to the appropriate source element and applying the corresponding edit. However, other types of edits define nesting and ordering relationships that must be taken into account. For example, many layout changes are applied hierarchically. In Figure 1(b), the designer may scale up the entire set of graphics for the Burj Khalifa and then separately adjust the vertical spacing and arrangement of the text elements in that column. Transferring this edit properly to all the buildings requires that we identify the analogous hierarchical structure for the other graphical elements in the design. In addition, the ordering between elements is important for temporal effects like animation. In Figure 1(a), the designer may want to apply animated entrance effects in a specific sequence to the set of elements for the Central Mediterranean region. Transferring this edit to the rest of the design requires grouping the elements for region and determining the appropriate animation order.

These various editing scenarios can be represented by specifying the desired nesting and ordering relationship amongst the source elements. Specifically, the input to our algorithm is the source elements represented as an ordered tree (*source tree*), and our goal is to organize the target elements into one or more ordered *target trees* that correspond to the source tree structure (Figure 4). Note that the problem becomes much simpler if each source element is allowed to match no more than one target element, or if we assume that the target elements are already organized into ordered trees that match the source tree. In such cases, finding the appropriate element-wise correspondence is sufficient. However, these assumptions are not realistic for most real-world design workflows. They either require users to manually select subsets of target elements to perform individual transfer operations, which is inconvenient for designs with many repeated components, or to arrange graphics into consistently ordered trees (which may differ per editing operation) ahead of time.

Thus, we propose an algorithm for computing the shared structure between source and target elements that does not limit the number of target elements or assume the presence of a consistent pre-defined structure in the design.

## 4 ALGORITHM

Our algorithm is composed of two main stages. First, we compute an element-wise correspondence by finding the best matching source element for each target element. Then, we compute a hierarchical clustering of the target elements and organize them into ordered target trees. Overall, our approach is closely inspired by Fisher et al. [4] and relies on graph kernels. While their method computes global similarity between entire 3D scenes, we need a detailed, structured correspondence between individual elements. This requires three new aspects in our approach.

- Finding an optimal element-wise match requires a similarity score for each source target-pair of elements (vs a single similarity score between two scenes from Fisher et al.), and an algorithm to find the best match using these scores. (Section 4.2.5)

- Determining the nesting and ordering that corresponds to the input edit operations requires clustering the target elements and inferring their order. (Section 4.3)

- We use different low-level kernels pertinent to comparing graphic designs. For example, while Fisher et al. use binary edge kernels by comparing edge types, we assign partial similarity by also considering the distance between elements. Such details are important to determine a correct match and

nesting for designs that may contain many elements that share identical relationships.

In the following, we use similar notation as Fisher et al. to illustrate how our method relates to and extends the previous work.

### 4.1 Relationship Graphs

Given the set of source elements $s$ and target elements $t$, we start by constructing *relationship graphs*, $G_s$ and $G_t$ for the source and target graphics, respectively. The nodes of the graph represent the graphical elements, and the edges specify relationships between those elements. We rely on the intuition that elements have certain relationships that characterize the structure of the design and make two designs more or less similar to each other. For example, in Figure 1(a), the text elements *'Central'*, *'Mediterranean'* and *'91,302'* are center-aligned with each other and all contained inside the orange circle element. The blue and green charts also contain analogous text and circle elements that share these relationships. We selected a number of prevalent relationships by observing real-world designs. Table 1 shows the list of the relationships and the process used to test for them. In general, the graph may contain multiple edges between a pair of nodes.

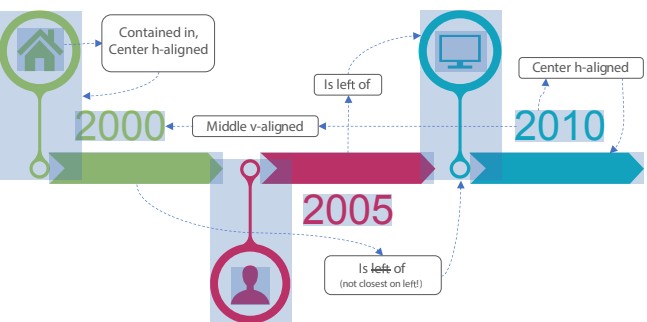

Figure 2: An example relationship graph for a set of elements. Only a subset of the edges are shown.

The tests are performed in the order listed. To eliminate redundancy, we encode at most one edge for a given category between any two elements. For example, if elements A and B are both center-aligned and left-aligned, we only encode the center-aligned relationship since that is the first test to be satisfied in the horizontal alignment category. Figure 2 is an example illustrating different edges in a relationship graph. Note that we do not encode grouping information from the designer. Grouping structure created during the authoring process is usually not a reliable indicator of the visual structure that informs most edit operations. Thus, we decided to disregard all such groups when constructing the relationship graphs.

### 4.2 Computing Element-wise Correspondence

After constructing source and target relationship graphs, we compute correspondences between their nodes. More specifically, for each target graph node $n_t \in G_t$, we find the closest matching source graph node $n_s \in G_s$ using a graph kernel-based approach inspired by Fisher et al. [4]. To apply the method, we first define separate kernels to compare individual nodes and edges across the two graphs.

#### 4.2.1 Node Kernel

The nodes in our relationship graph represent individual graphical elements, with several properties such as type, shape, size, and style attributes. The node kernel is a combination of several functions, each of which takes as input two nodes and computes the similarity of different features of the nodes. Each function described below is

| Category | Edge | Test |
|---|---|---|
| Intersection | Overlay | Element A and element B have the same bounding boxes. |
| | Contained in | Element A is contained in Element B if A's bounding box is inside the B's bounding box. |
| | Overlap | Element A overlaps element B if their bounding boxes intersect |
| Z-Order | Z-Above / Z-Below | Element A is Z-Above (Z-Below) element B if element A and B overlap, and A's z-order is higher (lower) than that of B. |
| Vertical alignment | Center / Left / Right | Similar to intersection relationships, alignment is computed on element bounding boxes, |
| Horizontal Alignment | Middle / Top / Bottom | but considers individual edges and centers of bounding boxes instead of areas. |
| Horizontal Adjacency | Left of / Right of | Element A is left-of element B if its bounding box is the closest bounding box to the left of B, |
| Vertical Adjacency | Above / Below | is within a threshold distance to B, and if the vertical range of their bounding boxes overlap.(*) |
| Style | Same Style | While there is a plethora of style attributes for each element, we use fill color and stroke style for non-text elements, and font style for text elements since these attributes are visually most apparent. |

Table 1: Edges encoded in relationship graphs. (*For threshold, we use $\frac{1}{2}$ (width of source graphics bounding box) for horizontal adjacency, and $\frac{1}{2}$ (height of source graphics bounding box) for vertical adjacency. If element A is left of multiple other elements, we only encode the relationship with the closest element. These constraints prevent edges between elements that are far apart relative to the size of the source graphics.)

constructed to be positive semi-definite and bounded between 0 (no similarity) and 1 (identical).

**Type Kernel** ($k_{type}$): Vector graphics elements are typically categorized as shapes (e.g., path, circle, rectangle), images, or text. In particular, most graphic design products and the SVG specification distinguish objects in this way. The type kernel returns 1 if two nodes have the same type (e.g., circle–circle), 0.5 if they are in the same category (e.g., circle and rectangle are both in the shapes category), and 0 otherwise (e.g,. circle–image).

**Size Kernel** ($k_{size}$): This function compares the bounding box size of two elements. It returns the area of the smaller bounding box divided by the area of the larger bounding box.

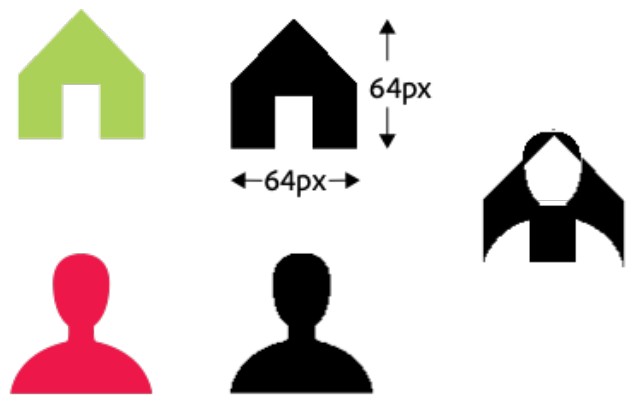

Figure 3: The element shape kernel, $k_{shape}$ computes the difference between the normalized bitmap images of the elements' silhouettes.

**Shape Kernel** ($k_{shape}$): We obtain the *normalized shape* (ignoring aspect ratio) of each element by taking its filled silhouette and scaling it into a 64 x 64 bitmap image (Figure 3). The element shape kernel returns the percentage image difference between two normalized shapes.

**Font Kernel** ($k_{font}$): For comparing two text elements, we consider their font style attributes. Specifically, we compare font-family, font-size, font-style (e.g., normal, italic) and font-weight (e.g., normal, bold). We return the percentage of style attributes that have equal values.

The final node kernel, $k_{node}$, is a weighted sum of the above kernels. Since many editing operations (e.g., changing font size, applying a character-wise animation effect) are non-transferable between text and shape elements, we separate text elements and non-text elements and only compare elements within the same category.

For comparing shape or image elements, we take into account type, size, and shape kernels.

$$
\begin{aligned}
k_{node}(n_s, n_t) = {} & \omega_{type}k_{type}(n_s, n_t) \\
& + \omega_{size}k_{size}(n_s, n_t) + \omega_{shape}k_{shape}(n_s, n_t)
\end{aligned}
\tag{1}
$$

For text elements, font style attributes are deemed more discriminatory than shape or size.

$$
k_{node}(n_s, n_t) = \omega_{type}k_{type}(n_s, n_t) + \omega_{font}k_{font}(n_s, n_t) \tag{2}
$$

If $n_s$ and $n_t$ are not in the same category, we assign a small constant ($k_{node} = 0.1$) instead. The weights, $\omega_{type}$, $\omega_{size}$, $\omega_{shape}$ and $\omega_{font}$, are defined per each source element. § 4.2.4 details how we compute these weights.

### 4.2.2 Edge Kernel

Next, we define an edge kernel to compute the similarity between a pair of edges that represent the relationship between two graphical elements. Each edge encodes a type of relationship (e.g., overlap, left-aligned). Based on our observation of real-world designs, we distinguish between *strong* edges and *regular* edges. Strong edges are highly discriminative relationships that tend to be preserved across design alterations. These include intersection and z-order relationships. All other edge relationships are deemed regular. In addition to the type ($\tau$), each edge also encodes the distance ($d$) between the two connected elements. Then, the kernel between two edges $e_s$ and $e_t$ with types $\tau_{e_s}$, $\tau_{e_t}$ and distances $d_{e_s}$, $d_{e_t}$ respectively is defined as:

$$
k_{edge}(e_s, e_t) = \omega_{\tau_{e_s}} c(\tau_{e_s}) \delta(\tau_{e_s}, \tau_{e_t}) \frac{min(d_{e_s}, d_{e_t})}{max(d_{e_s}, d_{e_t})} \tag{3}
$$

where $\delta$ is a Kronecker delta function which returns whether the two edges types $\tau_{e_s}$ and $\tau_{e_t}$ are identical. $c(t_e)$ is 2.5 if $\tau_e$ is a strong edge, and 1 if it is a regular edge. In our work, we use centroid distance as a gross approximation for distances between elements, but other distance metrics such as Hausdorff distance can also be used. Again, $\omega_{\tau_{e_s}}$ is a weight factor that is computed per source element and per edge type. See §4.2.4 for details.

### 4.2.3 Graph Walk Kernel

Using the node and edge kernels we define a graph walk kernel to compute the similarity between nodes in two graphs. A walk of length $p$ on a graph is an ordered set of $p$ nodes on the graph along with a set of $p-1$ edges that connect this node set together. We exclude walks that contain a cycle. Let $W_G^p(n)$ be the set of all walks of length $p$ starting at node $n$ in a graph $G$. To compare nodes $n_s$

and $n_t$ in relationship graphs $G_s$ and $G_t$ respectively we define the $p$-th order rooted walk graph kernel $k_R^P$.

$$k_R^p(G_s, G_t, n_s, n_t) =$$
$$\sum_{W_{G_s}^p(n_s), W_{G_t}^p(n_t)} k_{node}(n_{s_p}, n_{t_p}) \prod_{i=1}^{p-1} k_{node}(n_{s_i}, n_{t_i}) k_{edge}(e_{s_i}, e_{t_i}) \quad (4)$$

The walk kernel compares nodes $n_s$ and $n_t$ by comparing all walks of length $p$ whose first node is $n_s$ against all walks of length $p$ whose first node is $n_t$. The similarity between a pair of walks is computed by comparing the nodes and edges that compose each walk using the node and edge kernels respectively.

Finally, the similarity of nodes $n_s$ and $n_t$ is defined by taking the sum of the average graph walk kernels for all walks up to length $p$

$$Sim(G_s, G_t, n_s, n_t) = \sum_p \frac{k_R^p(G_s, G_t, n_s, n_t)}{|W_{G_s}^p(n_s)||W_{G_t}^p(n_t)|} \quad (5)$$

where $|W_G^p(n)|$ is the number of all walks of length $p$ starting at node $n$ in a graph $G$.

### 4.2.4 Kernel Weights

The node and edge kernels in Equations 1 - 3 compare different features (e.g., shape, style, layout) of the source and target graphics. The weights applied to these kernels represent the importance of each feature in determining correspondence. It is not possible to assign globally meaningful weights because the discriminative power of each feature depends on the specific design and even specific elements within the design. For example, in Table 2, D4, 7 out of the 8 source elements have the same color, green. So, color is not a very discriminative feature for these elements. On the other hand, the circle element which contains the symbol has a unique pastel color within the source set. For this element, color is a highly discriminative feature. We assume that features that are highly discriminative within the source graphics will also be important when comparing to the target graphics. Based on this assumption, we determine a unique set of kernel weights for each element in the source graphics, $s$, as follows.

**Node kernel weights:** For each element $n_{s_i} \in s$, we compute the node feature kernel between $n_{s_i}$ and every other element $n_{s_j} \in s$. The weight $\omega_{feature}$ is inversely proportional to the average feature kernel value.

$$\omega_{feature}(n_{s_i}) = 1.0 - \frac{\sum_{n_{s_j} \in s, j \neq i} k_{feature}(n_{s_i}, n_{s_j})}{|s| - 1} \quad (6)$$

where $|s|$ is the number of elements in the source graphics. A high average feature kernel value means many elements within the source graphics share the same feature, so the feature is less discriminative and vice versa.

**Edge kernel weights:** The weights for the edge kernel is defined for each source element, $n_{s_i}$, and for each edge type, $\tau_e$.

$$\omega_{\tau_e}(n_{s_i}) = 1 - \frac{\sum_{e_i \in E_{n_{s_i}}} \delta(\tau_{e_i}, \tau_e)}{|E_{n_{s_i}}|} \quad (7)$$

where $E_{n_{s_i}}$ is the set of all edges from node $n_{s_i}$. The numerator counts the number of edges in $E_{n_{s_i}}$ that has type $\tau_e$. If element $n_{s_i}$ has many edges of a given type, that edge type or relationship is less discriminatory for $n_{s_i}$ and vice versa. In §6.2, we conduct an ablation experiment, where we replace these weights with uniform weights.

### 4.2.5 Element-wise Correspondence

Given the pairwise similarity score between the source and target elements (Equation 5), a straightforward approach for finding an element-wise correspondence would be to match each target element to the source element that has the highest similarity score. The downside of this approach is that it is sensitive to small differences in the similarity score. Instead, we take an iterative approach that looks for *confident* matches and utilizes these matches to update the similarity scores of other pairs of elements. Source element $n_{s_i}$ and target element $n_{t_j}$ is a confident match if

$$Sim(G_s, G_t, n_{s_i}, n_{t_j}) \gg Sim(G_s, G_t, n_s, n_{t_j}) \quad \forall n_s \in s, n_s \neq n_{s_i} \quad (8)$$

that is, if the similarity score of $(n_{s_i}, n_{t_j})$ is much greater than the similarity score of $n_{t_j}$ with any other source elements.

Once we identify the confident matches, we use them as *anchors* to re-compute all other pair-wise similarity scores. First, we update the node kernels in Equations 1-2. Since we are confident that $(n_{s_i}, n_{t_j})$ is a good match, we boost their node kernels:

$$k_{node}(n_{s_i}, n_{t_j}) = 2.5 \quad \text{if } (n_{s_i}, n_{t_j}) \text{ is a confident match} \quad (9)$$

Since our goal is to match each target element to exactly one source element, if $(n_{s_i}, n_{t_j})$ is a confident match $n_{t_j}$ should not match any other source element $n_s \neq n_{s_i}$. Therefore, we discount these node kernels:

$$k_{node}(n_s, n_{t_j}) = 0.1 \quad \text{if } (n_{s_i}, n_{t_j}) \text{ is a confident match and } n_s \neq n_{s_i}$$
$$(10)$$

All other node kernels remain the same. The similarity scores are re-computed using the original Equation 5 with the updated node kernels. We iterate the steps of identifying confident matches and re-computing the similarity scores until all target elements are matched to a source element or until we reach a set maximum number of iterations. At this point, any remaining target elements are matched with the most similar source element according to the most recently updated similarity scores.

## 4.3 Computing Ordered Target Trees

The previous stage of the algorithm finds the closest matching source element for each target element. As noted earlier, this correspondence is enough to transfer isolated attributes that do not depend on the nesting structure or order of the target elements. However, for many other edits (e.g., layout, animation), transferring changes requires computing ordered target trees that are consistent with the source tree.

### 4.3.1 Hierarchical Clustering

We start by computing a hierarchical clustering of the target elements that matches the structure of the source tree. For example, consider Figure 4(a). The designer may apply edits to the yellow section of the bulb (source), then select the rest of the sections at once (target) to transfer the edit. In this case, there are 4 top-level clusters within the target tree, each corresponding to the red, purple, navy, and blue sections of the bulb. If the designer grouped and rotated the flashlight and the light rays in the source graphics by 45 degrees to get a tilted effect, we need the nesting of the neckties shown in Figure 4(b), T1 to transfer that edit properly (since we would want to rotate each necktie individually as opposed to all of them in a block or two parts of the neckties separately). To obtain these clusters, we use agglomerative nesting (AGNES), a standard bottom-up clustering method that takes as input a similarity matrix ($D$) and the number of desired clusters ($k$), and iteratively merges the closest pair of clusters to generate a hierarchy [14].

The key aspect of our approach is how we define the similarity matrix, which measures how likely a pair of target elements should

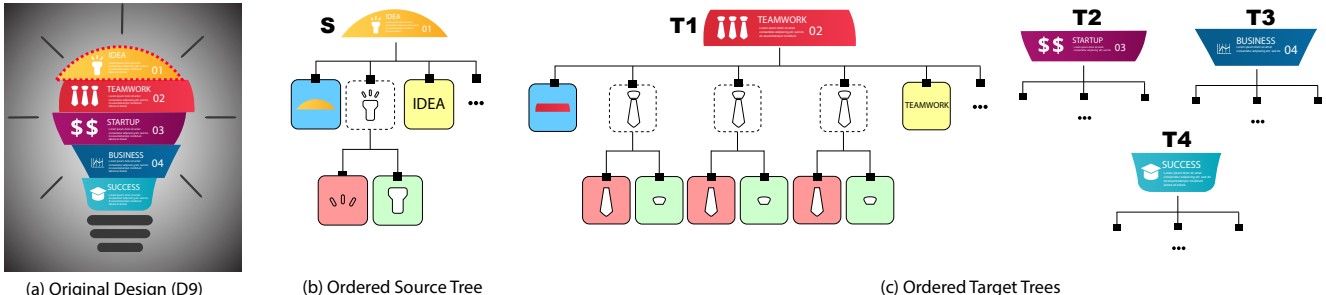

(a) Original Design (D9)          (b) Ordered Source Tree                    (c) Ordered Target Trees

Figure 4: Example of ordered source and target trees. Node colors indicate element-wise correspondence. Multiple instances of matching target graphics are represented as a nested target tree. Here, the entire source graphics (top slice of the bulb), matches 4 target instances, represented by sub-trees T1, T2, and T3. Within T1, there are 3 instances/sub-trees of the necktie symbol which corresponds to the flashlight symbol in the source. Our algorithm takes as input an ordered source tree and outputs an ordered target tree.

be clustered together. We rely on the intuition that the relationship between two target elements that belong to the same cluster would be similar to the relationship between their corresponding source elements. For example, in Figure 4, consider the yellow cross-section and the flashlight icon in the source graphics, and their corresponding elements in the target graphics (red, purple, navy, blue cross-sections and necktie, dollar sign, graph, and graduation hat icons). The yellow cross-section contains the flashlight. Likewise, the red cross-section contains neckties and these elements should be grouped together. On the other hand, although the dollar signs also correspond to the flashlight, the dollars signs are *not* contained in the red cross-section, and these should be grouped separately. We measure the similarity of relationships between pairs of elements, again using the graph walk kernel. From the relationship graph defined in §4.1, the relationship between two elements $n$ and $n'$ is represented by $W_G^p(n, n')$, the set of all walks of length $p$ whose first node is $n$ and whose last node is $n'$. Then, the similarity of target elements $n_t$ and $n_t'$ with corresponding source elements $n_s$ and $n_s'$ is defined as:

$$k_R^p(G_s, G_t, n_t, n_t', n_s, n_s') =$$

$$\sum_{\substack{W_{G_s}^p(n_s, n_s') \\ W_{G_t}^p(n_t, n_t')}} k_{node}(n_s, n_t) \prod_{i=1}^{p-1} k_{node}(n_{s_i}, n_{t_i}) k_{edge}(e_{s_i}, e_{t_i})$$

$$(11)$$

This equation is equivalent to Equation 4, except here we compare walks between a fixed source and destination node. That is, we compare all walks of length $p$ starting at $n_t$ and ending at $n_t'$ against all walks of length $p$ starting at $n_s$ and ending at $n_s'$. We take the sum of the average graph walk kernels for all walks of length up to $p$. For the bottom-up clustering method, to compare distances between two clusters of elements, we use the average distance between all pairs of elements from each cluster.

We use a simple heuristic to determine the number of clusters, $k$. For each source element, we count the number of matched target elements and take the mode of this value. We apply this heuristic recursively to determine the number of clusters at each level. We experimented with more complex methods such as using the spectral gap of the Laplacian matrix or putting a threshold on the maximum distance between two clusters to be merged. However, we found that the simpler approach worked better in most cases, even with variations in element cardinality between the source and target graphics.

### 4.3.2 Ordering

Once we obtain the target tree, the final step is to determine the ordering between the sub-trees, which translates to the ordering

of the elements. The ordering between sub-trees depends heavily on the global structure of the design (e.g., radial design *vs* linear design), the semantics of the content (e.g., graphics that represent chronological events), and the user's intent (e.g., presenting things in chronological order *vs* in reverse chronological order). Instead of trying to infer these factors, we rely on a simple heuristic based on natural reading order. We order the sub-trees from left to right, then top to bottom using their bounding box centers.

## 5 RESULTS

To test the effectiveness of our algorithm, we collected a set of 25 vector graphic designs. The designs included infographics, presentation templates, and UI layouts. We collected designs that contained several repeating structures that could benefit from bulk editing. We selected the source and target graphics for each design. Then, we manually coded the type of variations (e.g., color, layout) applied between the source and target graphics. More variations imply greater differences between the source and target graphics, making the correspondence more difficult to compute. Note that we did not use this test dataset to develop our algorithm.

For each design, we manually specified an ordered source tree which would correspond to a specific set of edits applied to the source elements and the corresponding ground truth set of ordered target trees. By default, we created shallow source trees (height of 1) for all designs. We also created deeper trees for a few examples (e.g., Figure 4. See supplementary material for more examples.) We chose the source element ordering such that applying entrance animation effects in that order would produce a plausible presentation. For example, text elements organized in paragraphs are ordered top-down, and container elements (e.g., the circle cross-sections in Figure 4) are succeeded by the interior graphics (e.g., icons and text). For other elements, we arbitrarily chose one specific ordering among many reasonable options. When constructing the ground truth target trees, we created a separate sub-tree for each set of target elements that correspond to an instance of the source graphics. For example, in Figure 4, the elements in of each cross-section constitute one target sub-tree. The ordering of elements within the lowest level sub-trees is determined by the ordering of the corresponding source element in the source tree. Higher-level sub-tree orderings are determined by the heuristic described in 4.3.2.

We evaluate the two stages of our algorithm separately. First, we evaluate the element-wise match between the source and target graphics. For the majority of target elements, the ground-truth match and clustering are visually apparent in the design. In cases where the variation between the source and target graphics make the match less clear, we choose a reasonable ground-truth match. For example, in Table 2 D5, there are six different *person* icons each of which consist of multiple vector graphics paths. Since, each path roughly

represents a body part (e.g., hair, face, neckline), we consider a match between equivalent parts of the symbol to be correct. On the other hand, the symbols in D4 do not have a clear semantic or visual correspondence. In this case, we consider a match between any target symbol path to any source symbol path as a correct match.

We evaluate the match for two different scenarios. In each design, the target graphics contains multiple sets of targets that match the source graphics. First, we evaluate the match on each target set separately (Separate). This would apply to the scenario when the user selects each target set separately and applies a transfer to each set one by one. Then, we evaluate the match between the source graphics and the entire target graphics, emulating the case when the user selects all target elements and applies the transfer at once (All). Table 2 reports the percentage of correctly matched target elements in each case.

For the hierarchical clustering and ordering encoded in the target trees, we also evaluate two scenarios. First, we compute the target trees given the match results from the All scenario, which may include incorrect matches (Default). Next, we compute the trees given the ground-truth match (Perfect Match). To quantify the correctness of the final result, we define an edit distance, $D_e$, that measures the difference between the ground-truth target tree and our computed target tree. To simplify this distance, we flatten both the ground-truth and computed trees into a sequence of target elements based on the ordering information encoded in the target trees. In these sequences, we label each target element with the corresponding source node, which allows us to identify errors in the computed element-wise matches (wrong label for a given target element) and incorrect ordering (wrong sequence of labels). Given a flattened ground-truth sequence $g$ and a computed sequence $r$, we define the edit distance as follows:

$$D_e(r,g) = D_m(r,g) + D_o(r,g) \qquad (12)$$

where the match edit distance, $D_m$, encodes differences in element-wise matches, and the order edit distance, $D_o$, encodes differences in the order of elements between $r$ and $g$. $D_m(r,g)$ is simply the total number of elements in the target that are matched to the wrong source element. It roughly represents the work required to fix all the incorrect matches. For the Default case, it is equal to the number of incorrect matches in All. For the Perfect Match case, it is equal to 0. The order edit distance is defined as:

$$D_o(r,g) = N - InOrder(r',g) \qquad (13)$$

where $N$ is the number of elements in the target graphics, and $r'$ is the result of correcting all matches in $r$. Note that $r'$ must be a permutation of $g$ since both contain the same set of target-to-source matches (potentially in different orders). $InOrder(r',g)$ is a function that returns the total length of all subsequences in $r'$ of length $\geq 2$ that exactly match a subsequence of $g$, minus $(N_o - 1)$, where $N_o$ is the number of matching subsequences. If $g$ and $r'$ are identical, $D_o(r',g) = 0$. $D_o$ is a variant of the transposition distance and roughly measures the number of operations required to correct the order of $r'$ to match $g$. Note that $D_e$ does not penalize $r$ for errors in the hierarchical structure of clusters as long as the result has the same ordering as $g$. In most cases, having the correct match and order will produce the desired visual result. Thus, $D_e$ attempts to approximate the number of correction operations needed to reach the desired ground truth solution. We report $D_e$ divided by the number of target elements, $N$. Note that creating $g$ manually would roughly correspond to $D_e = N$, since the user could just visit all the target elements in the correct order and assign each target element to the appropriate source element (or, equivalently, apply the desired edit).

**Element-wise Correspondence.** Table 2 shows the result for a subset of the designs that we tested. The full set of results is included in the supplementary material. The source graphic is highlighted with a red outline. The rest of the graphics minus the greyed-out background is the target graphics. We obtain close to perfect element-wise matches even when the target graphics has multiple types of variations from the source graphics. The match performance was comparable across the Separate and All scenarios, with only slightly better accuracy for the Separate case. This means that the user can transfer edits from the source graphic to multiple sets of target graphics at once without having to select each target set individually, which is especially tedious when there are many elements in a complex layout.

**Ordered Target Trees.** The accuracy of the ordered target trees varied widely across the designs. In order to obtain a perfect result, we must infer the correct match as well as the correct number of clusters ($k$). Error in either of these steps can have a big impact on the edit distance, $D_e$. For example, our algorithm computed a perfect element-wise match for D1. However, because the target graphics contained much fewer elements than the source graphics, our simple heuristic incorrectly inferred that $k = 1$. This led to a large $D_e$ since many elements were out of sequence. On the other hand, for D7, our heuristic correctly inferred that $k = 3$, and in fact, the clustering algorithm accurately identified elements belonging to each target set, in this case, the individual *profiles*. However, the relatively poor element-wise match mixed up the ordering of the elements within each target subtree, resulting in a larger $D_e$. Manually correcting the match (Perfect Match) and recomputing the target trees produced a perfect result ($D_e = 0$). In general, if the ordering error is due to the incorrect element-wise match, correcting the match will improve the accuracy of the resulting target trees. However, if the clustering of targets itself is wrong, correcting the match can alleviate the error only partially. In section 6, we also compare the results given a ground-truth $k$.

**Editing Applications.** Our algorithm for computing correspondences and ordered target trees supports a variety of edit transfer scenarios. To demonstrate this application, we implemented a prototype animation editor for SVG graphics that uses our automatic computation to transfer animation patterns from source to target elements. As noted earlier, transferring animation effects is uniquely challenging because they often involve both temporal and hierarchical structure.

In our interface, shown in the supplementary video, the user starts by applying animation effects to the source elements. The animation sequence itself specifies a nesting and ordering structure of the source graphics without requiring additional user input. Then, the user can select the target graphics and apply a transfer. Our system propagates the animation effects to the target elements by inferring the nesting and order of the target elements. The run-time of our algorithm depends on the number of the source and target elements. For the designs presented in the test set, our algorithm runs in interactive real-time.

Figure 1 shows other possible edit transfer scenarios. We created these examples by computing correspondences and ordered target trees for each set of source elements and then manually applying the corresponding edits to the target elements. In this process, we did not correct or modify the automatic output of our algorithm.

## 6 ABLATION EXPERIMENTS

To further evaluate the impact of different aspects of our algorithm, we conducted ablation experiments by removing key parts or our method or replacing them with simpler baselines.

### 6.1 Removing Edge Kernels

The graph walk kernel defined in Equation 4 compares two nodes $n_s$ and $n_t$ by comparing the similarity of their respective relationships with neighboring nodes. The walk length, $p$, determines the size

| Design | | # Elements | | Variations | | Match Score | | Tree Score ($D_e/N$) | |
| --- | --- | --- | --- | --- | --- | --- | --- | --- | --- |
| | | Source | Target($N$) | Count | Type | Separate | All | Default | Perfect Match |
| Avg | | 11.0 | 36.6 | 4.23 | | 0.95 | 0.95 | 0.31 | 0.15 |
| D1 |  | 17 | 21 | 2 | Text Cardinality | 1.00 | 1.00 | 0.81 ($k=1$) | - |
| D2 |  | 10 | 50 | 3 | Color Shape Size | 1.00 | 1.00 | 0.06 ($k=6$) | - |
| D3 |  | 9 | 36 | 4 | Cardinality Shape Size Layout Text | 1.00 | 0.97 | 0.03 ($k=4$) | 0 ($k=4$) |
| D4 |  | 8 | 33 | 5 | Cardinality Color Layout Shape Text | 1.00 | 1.00 | 0 ($k=4$) | - |
| D5 |  | 16 | 60 | 5 | Cardinality Color Layout Shape Size | 0.90 | 0.90 | 0.50 ($k=5$) | 0.08 ($k=5$) |
| D6 |  | 10 | 33 | 5 | Cardinality Color Size Text Layout | 1.00 | 1.00 | 0 ($k=2$) | - |
| D7 |  | 7 | 23 | 6 | Cardinality Layout Shape Size Text Type | 0.78 | 0.78 | 0.65 ($k=3$) | 0 |

Table 2: Representative examples from our test dataset. For each design, we evaluate the element-wise correspondence and clustering separately. Please refer to supplementary material for full results.

of the neighborhood to consider. In our implementation, we experimentally determined that $p = 1$ obtained satisfactory results. That is, considering only immediate neighbors was enough to predict good element-wise correspondences. We compare this approach to $p = 0$, where we disregard the relationships between the nodes and only use the node kernels to compute element matches. Table 3 (column `Node Kernels Only`) shows the result.

| Design | Match | | | |
|---|---|---|---|---|
| | Ours $p = 1$ | Node Kernel Only $p = 0$ | Uniform $\omega$ $p = 1$ | Greedy Match |
| Average | 0.95 | 0.82 | 0.93 | 0.93 |
| D1 | 1.00 | 0.95 | 1.00 | 0.83 |
| D2 | 1.00 | 0.76 | 1.00 | 1.00 |
| D3 | 0.94 | 0.80 | 0.94 | 0.97 |
| D4 | 1.00 | 0.76 | 1.00 | 1.00 |
| D5 | 0.90 | 0.78 | 0.86 | 0.87 |
| D6 | 1.00 | 0.94 | 0.97 | 1.00 |
| D7 | 0.78 | 0.78 | 0.78 | 0.78 |

Table 3: Ablation experiments for element-wise matching.

In the majority of cases, removing edge kernels and only considering node kernels produced worse results. This was especially true for designs that contained many elements that *looked alike*, with similar shapes and sizes. For example, in D3, the shorter target green bars get matched to the source blue bar because they have similar sizes. Likewise, in D6, each horizontal bar in the target graphics get matched to the source bar with the closest size, rather than being matched according to their relative positions. A particularly bad failure case is shown in Figure 5, D10 (Ours = 1.0 vs `Node Kernel Only` = 0.47), where the shadow consists of multiple circle elements with the same shape and size. In this case, without the z-order or relative positioning information, it is challenging to get a correct pair-wise correspondence between these circles. This experiment demonstrates that inter-element relationships are critical for discerning element-wise correspondences in graphic designs.

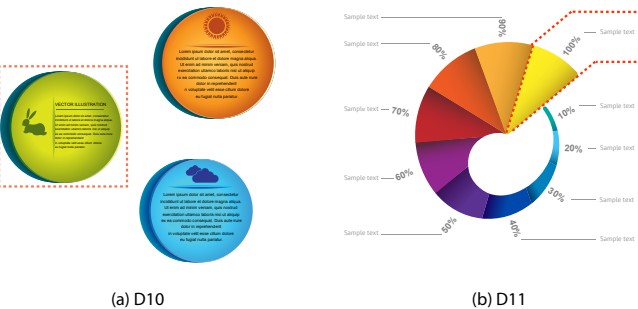

(a) D10                    (b) D11

Figure 5: (a) Design with many elements that have a similar appearance. The shadows consist of multiple circle elements with the same shape and size. Inter-element relationships (e.g., z-order) are critical for discerning correspondence between such designs. (b) Depending on the design, some features are more discriminatory than others. In D11 element type and font style attributes are more powerful features than relative positioning between the elements.

## 6.2 Uniform Kernel Weights

In §4.2.4, we describe a method for determining the importance and thus the weights ($\omega$) of each feature kernel. We evaluate the effectiveness of these weights by replacing them with uniform weights and comparing the results. Interestingly, in most cases, kernel weights did not have a significant impact on the performance. In a few cases,

adaptive weights (ours) produced better matches compared to uniform weights. For instance, in D5, all the symbol paths have the same type (*path*) so we put a small weight on the type kernel, and higher weights on other features such as the positions of the paths relative to each other. This helps to differentiate the subtle difference between these paths. Adaptive weights are also useful for matching elements where some features are much more powerful than others. For example, in Figure 5, D11 (`Ours` = 0.97 vs. `Uniform Weights` = 0.86), the type and font style attributes are much more powerful than the shape or layout relationship features. Still, for most designs, there are many features with discriminatory power, and replacing $\omega$s with a uniform weight produces as good a match as our previous result.

## 6.3 Greedy Matching

In §4.2.5, we describe an iterative method by which we first match *confident* pairs of nodes, and then use these matches to iteratively refine the similarity score of other pairs of nodes. We compare this approach to a greedy algorithm, whereby we simply match each target node to the source node with the highest similarity score. Table 3 column `Greedy Match` shows the result. For some designs, the greedy method produced as good a match as our iterative method. However, for other designs (e.g., D1 and D5), the greedy method performed worse. These were designs where a target element had several closely similar source elements. For example, in the UI design shown in D1, the different text elements for the user input fields are all very similar.. In D5 the symbol paths, as well as the solid blocks with 4 sides, are all very similar to each other. In these scenarios, using the more *confident* matches to refine the similarity scores helped improve the pair-wise match.

## 6.4 Clustering using Element Positions

The second stage of our algorithm (§4.3) uses hierarchical clustering of the target elements to compute target trees. As noted in §4.3.1, a key insight of our method is that the relationship between two target elements that belong to the same cluster should be similar to the relationship between their corresponding source elements. As a result, we use the graph walk kernel to analyze every pair of target elements and populate the similarity matrix used by the clustering procedure. To evaluate the importance of this insight, we compare our approach to a simpler heuristic that defines the similarity between every pair of target elements as their Euclidean distance (i.e., closer means more similar). We approximate element positions using the bounding box centroids. For both methods, we provide the ground-truth element-wise correspondences and the correct number of clusters, $k$.

Table 4 reports the results of the comparison. Using centroid distance produces worse clusters, especially when the desired target clusters are close to each other and arranged in a nonlinear layout (e.g., D2, D5). Even for designs with relatively simple layouts like D3, where the target clusters are visually separated from each other, the difference between the vertical and horizontal spacing coupled with the *tall* aspect ratio of some of the elements makes it challenging to estimate the correct clustering using only element centroids. In general, users can select an arbitrary arrangement of elements as the source graphics (not just ones that are geometrically close to each other). The graph walk kernel distance is better suited to handle these cases by producing clusters that reflect the original arrangement of the source graphics, as shown in the synthetic toy example, D8.

## 7 LIMITATIONS AND FUTURE WORK

**User Interface.** Our algorithm for computing correspondence trees supports a variety of interactive edit scenarios for static and animated graphics, but we have only implemented this functionality in a prototype interface for transferring animation effects. Future

| Design | Grouping ($D_e/N$) | |
| --- | --- | --- |
| | Ours Graph Kernel Distance | Centroid Distance |
| Avg | 0.02 | 0.09 |
| D1 | 0 | |
| D2 | 0.06 | 0.36 |
| D3 | 0 | 0.14 |
| D4 | 0 | |
| D5 | 0.08 | 0.12 |
| D6 | 0 | |
| D7 | 0 | |
| D8 | 0* | 0.35 |

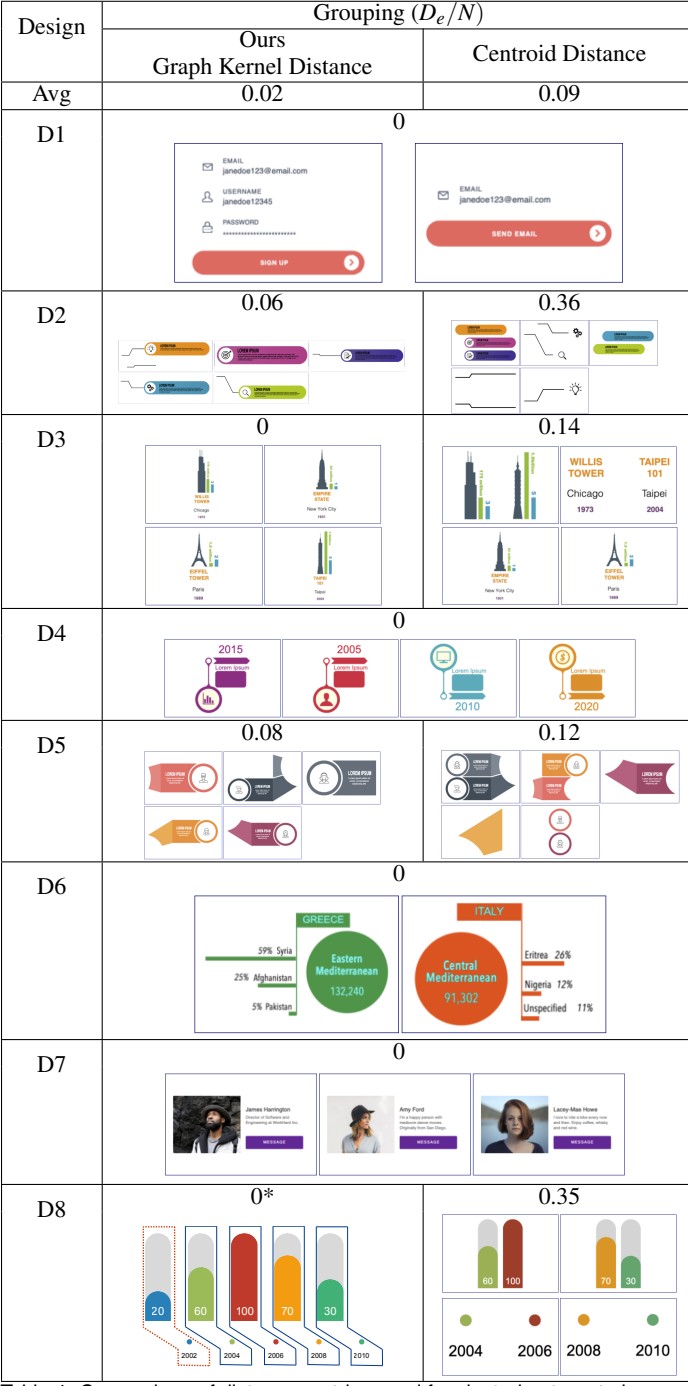

Table 4: Comparison of distance metrics used for clustering target elements (Graph Walk Kernel *vs* Centroid distance). The graph walk kernel distance produces clusters that reflects the original arrangement of the source graphics. *D8: red outline indicates source graphics, blue indicates target clusters.

work could address the design and evaluation of appropriate editing widgets and interactions to use our algorithm effectively.

**Edit History.** Future work could use the document's editing history to inform and refine the matching and clustering operations, potentially providing more accurate results based on the temporal order of how elements were added or changed. For example, knowing which elements were copy-pasted or which elements were jointly modified could provide hints about matching elements. This edit history usage is possible for both the source and target groups.

**Manual Corrections.** Using manual user corrections is another way to improve the algorithm's performance. When the user fixes an incorrect match, we could use this ground-truth match as a *confident* match to recompute the graph walk kernels or to infer better weights for the kernels. If there are multiple output errors, we could potentially reduce the number of required manual corrections by propagating the user's correction to other matching trees. In general, since the type and power of discriminatory features varies by design and user intent, learning from user inputs is an interesting avenue for future work.

**Limitations.** Our algorithm is designed to handle many variation types between the source and target graphics. Still, large stylistic, geometrical, and structural variations tend to produce incorrect matching and clustering results. Clustering is more prone to error because of its sensitivity to both the number of clusters, *k*, and the match results. We use a simple heuristic to determine the number of target clusters which works for many cases. This method can also fail easily when there are extraneous elements on the same canvas. Messier designs that don't include enough layer or grouping information to cleanly select the source and target graphics challenge our method. Future work to cleanly segment relevant clusters from complex designs could greatly extend the practical usage of our algorithm. Alternatively, users could provide the ground-truth *k* as input but this becomes tedious if the desired target tree is deeply nested and each sub-tree requires a different *k* value.

## 8 CONCLUSION

We present an approach to help designers apply consistent edits across multiple sets of elements. Our method allows users to select an arbitrary set of source elements, apply the desired edits, and then automatically transfer the edits to a collection of target elements. Our algorithm retroactively infers the shared structure between the source and target elements to find the correspondence between them. Our approach can be applied to any existing design without manual annotation or explicit structuring. It is flexible enough to accommodate common variations between the source and target graphics. Finally, it generalizes to different types of editing operations such as style transfer, layout adjustments, or applying animation effects. We demonstrate our algorithm on a range of real-world designs and show how our approach can facilitate editing workflows.

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
