# OpenReview forum: "Multi-level Correspondence via Graph Kernels for Editing Vector Graphics Designs"
_graphicsinterface.org/Graphics_Interface/2021/Conference — GI 2021_

### Official Review · AnonReviewer2 · 2020-12-29
**Review of Multi-level Correspondence via Graph Kernels for Editing Vector Graphics Designs**

**Rating:** 6
**Confidence:** 3

**Review:**

This paper presents a method for identifying correspondences between elements in a vector graphic.  Individual objects such as paths and text can be compared in terms of shape, size, and rendering properties.  More complicated compound objects can then be compared by evaluating the similarity of the individual components, as well as the similarities of the geometric relationships between those components.  Once compounds are identified based on a source compound identified by a user, edits can be transferred across in bulk, allowing for editing operations at a very high semantic level.  The method is based on graph kernels.  The paper includes lots of interesting use cases and an ablation study to demonstrate the utility of the various heuristics that make up the similarity computations.

I like this idea a lot.  Generally speaking I'm excited about any research that attempts to infer semantic content from raw vector graphics ("path soup"), in order to support content-aware manipulations.  The results suggest that the technique presented here is at least somewhat useful and reliable, though it's hard to tell how well that would scale to a general graphic design context.  I think the paper does a good job connecting to past work.  It might also be useful to cite these two papers:

	http://www.gilbertbernstein.com/project_lillicon.html
	https://research.adobe.com/publication/dynamic-planar-map-illustration/

Both involve some measure of inferring the structure of vector graphics images in order to process them intelligently.

Overall I found the paper to be both long and confusing.  I'm left concerned.  It seems like there may simply be too much to say in the form factor of a GI paper.  And even with all the details that are presented, I'm still not sure I understand the algorithm in full.  I'm also not sure I understand when it will perform well or poorly.

Some of this undoubtedly stems from my lack of familiarity with the techniques used.  Given that we're in a Graphics/HCI venue, and given that graph kernels are somewhat exotic, it may be worthwhile to introduce them more gently in this paper with an extra paragraph or two of text, instead of simply leaning on repeated references to Fisher, Savva, and Hanrahan [2].  (If nothing else, "kernel" is already a confusingly overloaded term in math and CS.)

It may even be possible to inject a new section around Section 3 that gives a high level view of the technique as part of an introduction to graph kernels, thereby preparing the reader to understand the meaning of the individual kernels that follow.

Beyond that, I have a few main concerns about the paper:

 * I'm not sure I understand what a typical user interaction might look like.  Does the user select the contents of the red box and blue box in Figure 1(a)?  Later, the paper seems to suggest that the user selects the Italy stats, and the system discovers the Hungary and Greece stats automatically from the entirety of the rest of the vector image.  Is that correct?  What if the rest of the image is very complex?  Is it still easy to find these correspondences within a large mass of undifferentiated paths?

 * On a related note, most of the examples are based on isolated infographics, where almost every path and text element is part of some cluster.  How does the system perform (in terms of speed and robustness) when the vector art contains a large amount of extraneous vector material that doesn't correspond to any part of the source?  Can it weed out all those distractions?

 * Has this technique been implemented in a real interactive tool, or in an offline prototype?  Is it fast enough to use in practice?  Given all the pairwise comparisons needed between elements, and need to aggregate kernels over all paths of a given length, I worry that such a tool couldn't run interactively.  The paper never discusses this.

 * In Section 6.4, measuring the distance between elements via the distance between their centroids isn't very convincing. I feel like it would make more sense to develop a distance metric that's a better match to human perception.  How about just measuring the smallest separation between the two objects (i.e., the minimum distance between any points on the two objects), or at least the shortest distance between the bounding boxes?

Here are a few smaller issues:

 * The writing could use a lot of improvement.  My biggest complaint is that one should never use a citation as a noun, as in "...[10] computes perceptual grouping of discrete patterns, and [3] encodes the structure...".  That's very hard to read.  There are also a number of typos and grammatical issues throughout the text that should be smoothed out with proofreading.  For example, on Page 2: "several design software", "loose their structural information", "data-formats", "3D scene comparisons".  "For For" in Table 1, etc., etc.

 * Section 4 might be more comprehensible if it were illustrated using a few abstract examples (i.e., simple paths and text rather than complete infographics).

 * Page 3 references Figure 4, but the figure is very far away in the paper.

 * What does "(not closest on left!)" mean in Figure 2?  Is that relationship explicitly encoded by the kernels?  Is it something else?

 * I don't understand the definition of "overlay" in Table 1.  Does it imply that A and B are identical?  That they have identical bounding boxes?

 * In what way are alignment relationships similar to intersection relationships?

 * It's weird that the shape kernel doesn't use symmetric difference.  I would expect Figure 3(c) to include the door of the house as black pixels, because the two images differ there.  It's also worth pointing out that the shape kernel is sensitive to rotation and reflection—a user may perceive rotated elements to be identical, but this kernel will miss that correspondence entirely.

 * The definition of k_type is a bit odd.  I'm not sure I completely understand what "same category" means.  More importantly, k_type is used as part of k_node, in a context where it will never be zero.  That is, k_node is defined conditionally in a way that doesn't require the full sophistication of k_type.  I'm not sure what to do about that.

 * I'm not sure I fully understand the motivation for the changing weights in 4.2.4.  Is that a new idea, or is it standard with graph kernel methods?  Perhaps an example could help.

 * The ablation study is interesting, and the results are somewhat convincing, but they still seem artificial.  I'd like to see a more "realistic" workflow being demonstrated if possible.  Maybe a video showing a typical interaction.  Maybe a few attempts to get professional graphic designers to use to the tool (see the Lillicon paper referenced above for a good example of this).

Overall I think the work is publishable, though it might conceivably be better off in a journal format, where it could be given more room to breathe.  As it is, I think it could use some careful rewriting to make the ideas clearer and the results more convincing.  I come down mildly positive about the work, but it'll be interesting to see what the other reviewers say about it.

---

### Official Review · AnonReviewer3 · 2021-01-10
**This paper describes an algorithm to transfer edits of vector graphics elements from a source to one or multiple target elements automatically. In order to facilitate this the structure of these elements is analysed and correspondences determined.**

**Rating:** 7
**Confidence:** 2

**Review:**

The authors propose to construct relationship graphs (nodes: graphics elements, edges: relationships, e.g. text alignment, color, font) and compute correspondences: a) element-wise, i.e. per node based on properties such as type, shape, size; b) edge-wise, i.e. for the relationship of two elements, e.g. overlap; and c) for graphs using a graph walk kernel based on a) and b). After matching elements simple transfers would be possible, but more complex edits such as layout operations, further require nesting structures or ordering of elements which is obtained from clustering (using AGNES) and heuristic ordering.

The proposed method (and both stages separately) is evaluated for several vector graphics, where the “ground-truth” has been constructed manually. Overall, I felt that the evaluation is quite extensive and basically demonstrated that the method works as suggested. Of course it would be very interesting to see the potential of making use of document edit history and user corrections (as suggested in the outlook), but this is clearly beyond the scope of this paper and future venue.

Overall the paper is well written and easy to follow; the previous work section is quite dense for someone not working in this field (i.e. no background/recap of basics), I can’t tell if there is prior work missing there.

---

### Official Review · AnonReviewer4 · 2021-01-14
**great problem and motivation; quantity of heuristics makes it hard to read**

**Rating:** 6
**Confidence:** 2

**Review:**

This paper tries to propagate editing of vector graphics elements to similar elements within the same design, using graph kernels to estimate similarity.  It is an ambitious and worthwhile goal, and the presented work seems to make some progress. Graph kernels are a plausible mechanism for similarity estimation in this context, even if some of the elements used here are oversimplified. (In particular, centroid distance struck me as a poor measure of proximity given the prevalence of non-circular elements in the examples, such as the buildings or the horizontal text boxes.)

I was intrigued by the paper but found it difficult to read, at least partly because the paper has a lot of different parts with no clear connection between them. The volume of heuristics (and subsequent testing of the heuristics) is high. I do not have a good solution to this, but the paper's length combined with the apparent immaturity of the method relative to its aims somewhat dampened my enthusiasm.

The similarity elements were chosen heuristically, based on observation of a set of reference designs. This is a reasonable way to proceed in early-stage work. I would hope that this could be improved on in future work, both by doing a more thorough survey and by appealing to general principles of graphic design.

Does the method work? Based on Figure 1, I expected to see some examples of design transfer being carried out on novel designs, but the paper doesn't have any of that. Can such edits happen in practice? How often would the method do something objectionable such that the user needs to correct it? Can it be fooled by extraneous elements, such as the map in Figure 1a? (The design may not include a convenient separation into layers.) There seems to be an implicit assumption about the simplicity of the design which may not always hold. In general, I thought there was a bit of a disconnect between the paper's aims and apparent contributions (taking Figure 1 as a preview of results) and then the actual results which concentrate on evaluating matches between manually-selected groupings and detected groupings. Should the paper be accepted, I would advocate being clearer about the paper's goals and contributions up front, with Figure 1 being emphasized less. (Or, if the authors do indeed have software that allows design transfers like those shown in Figure 1, they should tell us about it.)

I appreciated the use of ablation experiments to explore the importance of different heuristic choices.

Minor points:

Citations are parentheticals and not words in a sentence. The paper incorrectly attempts to use them as words, with usages like the following: "Our algorithm is directly inspired by [2]..." This is obnoxious and should be replaced by something like "directly inspired by the work of Fisher et al."

Reference 8 seems to be messed up, with the lowercase "i" and the unclear chapter tag.

---

### Official Review · AnonReviewer1 · 2021-01-15
**This work is a solid contribution to the problem of optimizing design edits. Specifically, this work proposes a robust algorithm for 2D vector graphics designs that automatically determines the design hierarchy and propagates edits without the need for human annotation.**

**Rating:** 7
**Confidence:** 4

**Review:**

PROS
Proposed algorithm is highly flexible with the ability to transfer design changes over multiple times of design relationships and has the benefit of not requiring user annotations.
Algorithm considers both design hierarchies with respect to design elements and animations.

CONS
Evaluation could be more detailed in comparing the proposed algorithm against the ablation studies with respect to pros and cons. Additionally, it would be interesting to evaluate the outputs with respect to the number of edits required by the designer in order to achieve the desired result. This latter point is touched on in future work as a way to improve the algorithm.

QUALITY
The algorithm tackles a commonly-used design space and design constraints when determining its hierarchy, making its utility promising. Additionally, the evaluation supports the computational complexity of the algorithm by demonstrating its performance on a variety of design types against variations of the algorithm from an ablation study.

CLARITY
The writing tends to have comma splices and choppy sentence transitions. Other than those points, the paper reads very well and communicates its problem and proposed solution clearly. Please see the subsection “Edit Notes” for more detail.
The paper layout causes the reader to repeatedly refer to figures and tables in other sections. For example, the text discusses Figure 4 after Figure 1 before Figure 2 or 3 are seen. Additionally, many examples refer back to Figure 1 on the first page.

ORIGINALITY
The idea of automatically propagating design edits has been done before, but this work focuses on removing as many constraints and burdens on the designer as possible. In other words, the problem is known and this solution makes forward progress.

SIGNIFICANCE
This tool would be very convenient for designers by removing the need for tedious edits. The evaluation is not complete enough to state how much effort this will save designers.

EDIT NOTES
Abstract First Sentence: To create graphic designs such as infographics, UI designs, and explanatory diagrams, designers often need to apply consistent…
Two sentences contain comma splices where no comma is needed:
Abstract: Our method does not require any explicit annotation, and can be applied to any existing design regardless of how it was created.
Abstract: We evaluate our algorithm on a range of real-world designs, and demonstrate how our approach can facilitate various editing scenarios.
First sentence of Overview: The input to our method is a set of source elements that the user has manually edited, and a set of target elements to which the user wants to transfer the edits.

---

### Meta-Review · Area_Chair1 · 2021-01-15

**Recommendation:** Accept
**Confidence:** 3

**Metareview:**

All reviewers felt that this submission was over the threshold for acceptance.  There was a bit of a tendency towards the middle: nobody offered crucial evidence that the paper cannot be published, and nobody championed it either. It's also worth noting that none of the reviewers offered an especially high confidence rating, which perhaps contributes to the middle-of-the-road scores. Putting all that information together, it seems reasonable to offer acceptance, with the hopes that the authors will carefully consider the full contents of the provided reviews and make appropriate changes to address them.

All reviewers appreciated the topic and its potential applications.  The technique seems to work well on the examples presented in the paper.  Some reviews mention the value of the ablation study.

There were also a few high-level concerns raised.  I believe the full reviews are all very useful, but I will single out some recurring themes:

 * The paper was difficult to read. It contained a lot of disconnected parts, and a large number of disparate heuristics. The writing could use improvement everywhere, taking note of annoyances like the use of citations as nouns (among other things).
 * The paper is long. The GI guidelines suggest "4-8 pages" as a starting point, and this paper was 11.  Might it be possible to offload some of the denser material into supplementary material or an appendix?
 * The paper must absolutely be clearer on the nature of the implementation.  Is this an interactive program?  Is it fast?  Does it work on general vector art?  How do edits happen in practice?  What does the user do if the algorithm chooses incorrectly?
 * The paper should clarify whether the algorithm can become confused by the presence of extraneous elements.
 * Evaluating inter-object distance via object centroids is hard to support.  An alternative would be welcome, or at least the suggestion of one as part of future work.

---

### Decision · Program_Chairs · 2021-01-16

Accept